# Depressive Symptoms in Middle-Aged and Elderly Women Are Associated with a Low Intake of Vitamin B6: A Cross-Sectional Study

**DOI:** 10.3390/nu12113437

**Published:** 2020-11-09

**Authors:** Tamami Odai, Masakazu Terauchi, Risa Suzuki, Kiyoko Kato, Asuka Hirose, Naoyuki Miyasaka

**Affiliations:** 1Department of Women’s Health, Tokyo Medical and Dental University, Tokyo 113-8510, Japan; odycrm@tmd.ac.jp (T.O.); kiyo.crm@tmd.ac.jp (K.K.); 2Department of Obstetrics and Gynecology, Tokyo Medical and Dental University, Tokyo 113-8510, Japan; 150612ms@tmd.ac.jp (R.S.); a-kacrm@tmd.ac.jp (A.H.); n.miyasaka.gyne@tmd.ac.jp (N.M.)

**Keywords:** depression, anxiety, menopause, mental disorder, pyridoxal 5′-phosphate

## Abstract

This study investigated the nutritional factors that are associated with anxiety and depressive symptoms in Japanese middle-aged and elderly women. We conducted a cross-sectional study with 289 study participants aged ≥40 years (mean age = 52.0 ± 6.9 years). Their dietary habits, menopausal status and symptoms, and varied background factors, such as body composition, lifestyle factors, and cardiovascular parameters, were assessed. Their anxiety and depressive symptoms were evaluated using the Hospital Anxiety and Depression Scale (HADS), where scores of 0–7 points, 8–10 points, and 11–21 points on either the anxiety or depression subscales were categorized as mild, moderate, and severe, respectively. The dietary consumption of nutrients was assessed using a brief self-administered diet history questionnaire. The relationships between the moderate-to-severe anxiety/depressive symptoms and the dietary intake of 43 major nutrients were investigated using multivariate logistic regression analyses. After adjusting for age, menopausal status, and the background factors that were significantly related to depressive symptoms, moderate and severe depression was significantly inversely associated with only vitamin B6 (adjusted odds ratio per 10 μg/MJ in vitamin B6 intake = 0.89, 95% confidence interval = 0.80–0.99). A higher intake of vitamin B6 could help relieve depressive symptoms for this population.

## 1. Introduction

Depression and anxiety are common mental disorders. According to the World Health Organization, the estimated number of people globally who have depression and anxiety has increased by 18.4% and 14.9%, respectively, over the last ten years, totaling more than 300 million and 250 million, respectively [1]. Women are more likely to have mental disorders than men [1]. Large fluctuations of the serum estrogen level in the premenstrual, postpartum, and perimenopausal periods contribute to mood changes [2,3]. An increased risk for mental disorders, including depression, is observed during the menopause transition [4,5]; additionally, the early menopause transition and early menopause are significant risks for depression, which indicates that a short duration of exposure to endogenous estrogens can increase the risk for late-life depression [6,7]. Estrogen plays neuroprotective, anti-depressive, and anti-anxiety roles via the regulation of serotonergic and noradrenergic systems [8]; however, the associations between psychological symptoms and absolute serum sex hormone levels are inconsistent [3,9]. Additionally, several reports have demonstrated the beneficial effects of hormone replacement therapy (HRT) on mood symptoms [10,11,12], while a randomized controlled trial covering a wide age range of postmenopausal women failed to show the effects of HRT on depressive disorders [13].

Furthermore, as there are increasing concerns about the side effects of HRT associated with cardiovascular diseases and hormone-sensitive cancers, such as breast, endometrial, and ovarian cancer, expectations for a complementary form of therapy are growing. Several reports have shown the association between specific nutrients, such as thiamine, folate, vitamin B6 and B12, zinc, and iron, and mental health [14,15]; however, the psychological effects of dietary intake of various nutrients remain largely unknown, especially in peri- and postmenopausal women. This study thus investigated the associations between anxiety/depressive symptoms and dietary consumption of nutrients in Japanese middle-aged and elderly women.

## 2. Materials and Methods

### 2.1. Study Population

This cross-sectional analysis of 700 Japanese women who had enrolled in the Systematic Health and Nutrition Education Program at the menopause clinic of the Tokyo Medical and Dental University was conducted based on the first-visit medical records from January 2009 to August 2017. All participants who registered in this program had sought medical attention for menopausal symptoms and had been provided improvement strategies based on the assessment of their physical and psychological health status and lifestyle.

We evaluated women’s anxiety/depressive symptoms using the Hospital Anxiety and Depression Scale (HADS) and their dietary habits using a brief self-administered diet history questionnaire (BDHQ). A total of 313 participants who had failed to complete the HADS and/or BDHQ, 46 participants who had been treated with estrogen, four participants who were aged <40 years, and 48 participants whose menopausal status was uncertain were excluded. For the remaining 289 women, the associations between anxiety/depressive symptoms and the dietary intake of nutrients were evaluated.

This study protocol was reviewed and approved by the Tokyo Medical and Dental University Review Board (approval number: 774) and written informed consent was obtained from all participants. The research was performed in accordance with the Declaration of Helsinki.

### 2.2. Measurements

#### 2.2.1. Menopausal Definitions

The women were defined as premenopausal if they had regular menstrual cycles, in the menopausal transition if they had menstruated within the past 12 months but had missed periods or had an irregular cycle in the past 3 months, and postmenopausal if they had not menstruated in the past 12 months. The postmenopausal women with surgically induced menopause via a hysterectomy and/or oophorectomy were excluded as their menopausal status was unclear.

#### 2.2.2. Physical Assessment

Participants’ body composition variables, including height, weight, body mass index, body fat percentage, fat mass, lean body mass, muscle mass, water mass, and basal metabolic rate, were measured using a bioimpedance analyzer (MC190-EM; Tanita, Tokyo, Japan). Waist and hip circumferences were measured to calculate the waist-to-hip ratio, and body temperature was measured with a thermometer. Resting energy expenditure was also determined based on the respiratory volume using an indirect calorimeter (Metavine-N VMB-005 N; Vine, Tokyo, Japan). Moreover, cardiovascular parameters, including systolic and diastolic blood pressure, heart rate, cardio-ankle vascular index, and ankle-brachial pressure index, were evaluated using a vascular screening system (VS-1000; Fukuda Denshi, Tokyo, Japan). Additionally, we utilized a physical fitness test to assess power, reaction time, and flexibility. Hand-grip strength was measured twice for each hand with a hand dynamometer (Yagami, Nagoya, Japan) and the mean value (kgf) was calculated using the larger value of the two measurements. The ruler drop test, which is an assessment of reaction time, was repeated seven times using a wooden ruler that was 60 cm in length and 110 g (Yagami, Nagoya, Japan). Participants fixed their arms on a desk and outstretched their fingers from the edge of the desk, while the bottom of the ruler was hung from between the thumb and index finger of an examiner; the participants then attempted to catch the ruler as quickly as possible when it was dropped. Where the participants gripped the ruler was evaluated, and the average reaction time (cm) was calculated from the remaining five values after the largest and smallest values were omitted. The sit-and-reach test for assessing flexibility was conducted with a reach box while sitting (Yagami, Nagoya, Japan).

#### 2.2.3. Lifestyle Factors

We investigated lifestyle characteristics, such as the frequency of smoking (none, fewer than 20 cigarettes per day, or 20 or more cigarettes per day), consumption of alcohol (never, sometimes, or daily), and caffeinated drinks (none, once or twice per day, or three or more times per day), and regular exercise habits (yes or no).

#### 2.2.4. Questionnaires

The HADS, which is a widely used screening instrument for anxiety and depression, was developed to evaluate the mental health of patients with somatic symptoms [16]. It consists of seven items across two subscales: anxiety and depression. Anxiety is assessed using feeling states: feeling tense, restless, or panicky; feeling something awful will happen; having worrying thoughts; feeling unable to relax; having butterflies in one’s stomach. Furthermore, depression is evaluated using anhedonia: unable to enjoy things, unable to laugh and see the funny side, not feeling cheerful, feeling slowed down, having lost interest in one’s appearance, unable to look forward to things, and unable to enjoy a book or TV. Each item is scored on a four-point Likert scale. Cut-off points to identify doubtful and definite cases for anxiety or depression are 8 and 11 points, respectively. In the present study, scores of 0–7 points, 8–10 points, and 11–21 points were classified as mild, moderate, and severe anxiety/depression, respectively.

The BDHQ, a short version of a self-administered diet history questionnaire that was developed in Japan, is composed of 77 questions and takes approximately 15 min to answer. The BDHQ was used to assess the intake frequency of 61 food items that are commonly consumed in Japan, mainly from the food list used in the National Health and Nutrition Survey of Japan [17], including beverages and seasonings, in the preceding month. Using an ad hoc computer algorithm for estimating the daily intake of nutrients and food after an adjustment for total calorie intake, the consumption of 96 nutrients and 58 food items was calculated. The estimated intake of nutrients and food items based on the BDHQ has previously been validated via a comparison with dietary records using a semi-weighted method [18,19]. In this study, we investigated the association between anxiety and depressive symptoms and 43 major nutrients with high validity (Appendix A).

We also evaluated the women’s health using the Menopausal Health-Related Quality of Life Questionnaire (MHR-QOL). The MHR-QOL, which is a modification of the Women’s Health Questionnaire and others [20,21,22], was developed and validated in our clinic [23,24,25,26] and comprises four categories: physical health, psychological health, life satisfaction, and social involvement (Appendix A). The items for physical and psychological symptoms are scored using a four-point Likert scale based on the symptom frequencies. According to the degree of agreement or disagreement, life satisfaction and social involvement were assessed using a two- or four-point Likert scale. We pooled the scores for somatic symptoms (6 items), vasomotor symptoms (2 items), insomnia symptoms (2 items), life satisfaction (5 items), and social involvement (12 items), to evaluate the score for each subcategory. A low score represented severe symptoms and low levels of life satisfaction and social engagement.

Lastly, we assessed psychotropic medication use, including hypnotics, anxiolytics, and antidepressants, based on medical interview responses.

### 2.3. Statistical Analysis

Continuous variables are presented as mean ± standard deviation. The required sample size was calculated as follows: 10 times the number of independent variables divided by the prevalence of anxiety or depressive symptoms, which were estimated to be 8 and 0.4, respectively, was 200. The comparison between groups was performed using the Kruskal–Wallis and the chi-squared tests. Cut-off points for the identification of multicollinearity were determined using a Pearson or Spearman correlation coefficient of *r* > 0.9. The nutrients and background factors associated with anxiety or depressive symptoms were evaluated using a stepwise regression procedure with a threshold of *p* = 0.1 and *p* = 0.05 for variable inclusion and exclusion, respectively. The analysis was conducted with a multivariate logistic regression model to clarify the relationships between the severity of anxiety and depressive symptoms and the selected nutrients. Significance was set at *p* < 0.05. All analyses were performed with GraphPad Prism version 5.02 (GraphPad Software, San Diego, CA, USA) and JMP version 12 (SAS Institute Inc, Cary, NC, USA).

## 3. Results

The participants’ (*n* = 289) mean age was 52.0 ± 6.9 years. The prevalence of mild, moderate, and severe anxiety was 54.0%, 28.7%, and 17.3%, respectively, and that of depression was 61.6%, 25.6%, and 12.8%, respectively. The participants’ background characteristics are shown in Table 1 and Table 2. The participants with severe anxiety/depressive symptoms had a low quality of life according to the assessment using the MHR-QOL and were less frequently engaged in exercise. Moreover, the participants with severe depressive symptoms were younger than those with mild symptoms. There was no significant difference in body composition or the physical fitness test between the three severity groups.

First, we assessed the daily intake of 43 nutrients; then, we investigated the nutritional intake, which differed significantly between the three anxiety/depression severity groups. The intake of 10 and 22 nutrients showed significant differences between the three groups of anxiety and depression severity, respectively (Table 3 and Appendix A). Similarly, the background factors related to the severity of anxiety and depressive symptoms were investigated. The factors that were significantly related to anxiety were insomnia and depression scores, while the factors associated with depression were life satisfaction, social involvement, and anxiety scores. Next, to identify the independent variables among these nutrients related to the severity of anxiety/depressive symptoms, a stepwise regression analysis was performed after eliminating multicollinearity. We found that the severity of both anxiety and depressive symptoms was only significantly associated with the intake of vitamin B6. Finally, a multivariate logistic regression analysis was performed to identify the independent relationships between the daily intake of vitamin B6 and moderate-to-severe anxiety/depressive symptoms. After adjusting for age, menopausal status (model 2), and background factors that were significantly related to the severity of anxiety/depressive symptoms (model 3), the intake of vitamin B6 was significantly associated with moderate-to-severe depression (model 2: adjusted odds ratio (AOR) per 10 μg/MJ in vitamin B6 intake = 0.91, 95% confidence interval (CI) = 0.85–0.97; model 3: AOR = 0.89, 95% CI = 0.80–0.99), while there was no significant relationship between the intake of vitamin B6 and the severity of anxiety after adjusting for the background variables (AOR = 0.97, 95% CI = 0.90–1.04; Table 4).

## 4. Discussion

In our cross-sectional analysis, the severity of depressive symptoms was significantly inversely associated with the dietary intake of vitamin B6 in Japanese middle-aged and elderly women. Vitamin B6, which comprises six chemical compounds (pyridoxine, pyridoxamine, pyridoxal, and their phosphorylated derivatives pyridoxine 5′-phosphate, pyridoxamine 5′-phosphate, and pyridoxal 5′-phosphate (PLP)), is richly contained in red pepper, garlic, nuts, fish, and meats. PLP, the most active form, serves as an enzymatic cofactor in more than 140 different biochemical reactions, such as those involving amino acids, neurotransmitters, heme biosynthesis, fatty acid metabolism, and glycogen breakdown [27,28].

It is well known that neurotransmitters, such as serotonin, dopamine, norepinephrine, γ-aminobutyric acid, and glutamate, play a critical role in the development of psychiatric disorders, and their receptors could be potential therapeutic targets for the treatment of psychoneurological symptoms. Abundant reports have shown that the dysregulation of monoamine systems contributes to anxiety and depressive disorders [29,30]. Serotonin is synthesized from tryptophan by PLP-dependent dopa decarboxylase, and dopamine and norepinephrine production are also required for the catalysis of PLP-dependent dopa decarboxylase. Decreased vitamin B6 (PLP) could be associated with monoamine depletion and impaired neurotransmission. Furthermore, the kynurenine pathway involving two PLP-dependent enzymes, which is a major tryptophan metabolic pathway, is associated with depression [31]. The disturbance of the balance between the neuroprotection and neurotoxicity of kynurenine pathway metabolites—i.e., kynurenines, such as kynurenic acid and quinolinic acid—plays a key role in the development of depression [32]. It is supposed that PLP-dependent kynureninase is more sensitive to PLP deficiency than is the PLP-dependent kynurenine aminotransferase; thus, PLP deficiency reduces kynureninase activity first [33,34]. Therefore, kynurenic acid, 3-hydroxykynurenine, and xanthurenic acid could increase, although, the changes in the plasma and urinary levels of kynurenines via vitamin B6 depletion are inconsistent [33,34,35]. Kynurenic acid inhibits N-methyl-D-aspartate receptors and alfa-acetylcholine receptors in the central nervous system [36,37], leading to a decline in the extracellular levels of acetylcholine, glutamate, and dopamine. Additionally, 3-hydroxykynurenine, which is a redox-active metabolite, is neurotoxic through generating reactive oxygen species and eventually induces apoptosis [38], while xanthurenic acid acts as a metabolic glutamate 2/3 receptor agonist, which could improve positive and negative symptoms in schizophrenia [39]. Vitamin B6 depletion might cause a dysregulated neurotransmitter system and neural dysfunction through imbalances of tryptophan metabolites via kynurenine.

There are several reports on the association between vitamin B6 and depression. Hvas and colleagues showed that low plasma levels of PLP were related to depressive symptoms in 140 study participants [40]. A seven-year longitudinal study of 3503 adults aged ≥65 years demonstrated that a higher total intake of vitamin B6 (dietary and supplementary intake) was related to a lower likelihood of depression [41]. Moreover, a higher dietary intake of vitamin B6 was associated with a lower incidence of depression among women in a three-year longitudinal study of 1793 adults aged ≥68 years [42]. Additionally, a few systematic reviews of the effects on mood of vitamin B6 alone, or a combinative intervention of vitamins and minerals, such as folate, vitamin B12, vitamin C, vitamin D, magnesium, calcium, and iron, on mood supported the idea that supplementation with B6 vitamins could relieve mood symptoms. For example, Williams and colleagues reported the beneficial effects of vitamin B6 supplementation on depression among premenopausal women [43]. Young and colleagues also revealed that the supplementation of B vitamins might alleviate mood symptoms in healthy adults and adults at risk for mental disorders [44].

In contrast, several randomized controlled trials failed to find significant effects due to a combinative intervention of B vitamins, including vitamin B6, on mood symptoms [45,46,47]. In the current study, the mean daily vitamin B6 intake was smaller than the recommended dietary allowance only in the severe depressive group [48], which might affect our results. Further studies should be conducted to determine the exact effects of vitamin B6 as an independent treatment.

The major limitations of our study were the relatively small sample size and uncertain causal relationship owing to its cross-sectional nature. It may not be appropriate to generalize our findings to a wider population. We did not investigate the serum levels of vitamin B6, although we estimated the daily intake of vitamin B6 using the BDHQ. Therefore, it was uncertain whether the severity of depression was related to serum vitamin B6 levels. Furthermore, the BDHQ, which is a method based on food recall to determine the frequency of food eaten, provided information only for the 61 listed foods and beverages. In addition, a potential contributor to mood, namely, the use of dietary supplements, was not assessed. The use of dietary supplements, such as vitamins (B1, B2, B6, C, and E) and minerals (calcium and iron), has been estimated at only 7.7% in Japanese women [49]. Nevertheless, the dietary intake in this study did not represent the total nutrient intake.

Nonetheless, our study has several strengths and novel features. As many as 43 nutrients and various background factors, including physical and psychological health status, life satisfaction, and social involvement, were analyzed simultaneously. Therefore, we found that the intake of vitamin B6 was independently associated with the severity of depressive symptoms. To the best of our knowledge, this is the first report on the relationship between the intake of vitamin B6 and depressive symptoms as a result of an analysis of various nutrients. 

In conclusion, moderate-to-severe depressive symptoms were associated with a lower dietary intake of vitamin B6 in Japanese middle-aged and elderly women. A higher intake of vitamin B6 could help relieve depressive symptoms in this population.

## Figures and Tables

**Table 1 nutrients-12-03437-t001:** The physical characteristics of all participants and comparison of these factors between women with mild, moderate, and severe anxiety and depressive symptoms.

Physical Characteristics	All Participants (*n* = 289)	Anxiety				Depression			
		Mild (*n* = 156)	Moderate (*n* = 83)	Severe (*n* = 50)	*p*-Value	Mild (*n* = 178)	Moderate (*n* = 74)	Severe (*n* = 37)	*p*-Value
Age (years)	52.0 (6.9)	54.3 (7.6)	52.6 (6.2)	52.3 (5.3)	0.279 ^a^	54.7 (7.5)	52.2 (5.6)	50.2 (4.3)	0.002 ^a^
Menopausal status (%)									
Pre/peri/postmenopausal	26.6/15.6/57.8	25.0/16.0/59.0	28.9/15.7/55.4	28.0/14.0/58.0	0.966 ^b^	25.8/13.5/60.7	28.4/17.6/54.0	27.0/21.6/51.4	0.665 ^b^
Body composition									
Height (cm)	157.0 (6.1)	156.6 (6.2)	157.5 (5.7)	155.8 (6.0)	0.206 ^a^	156.4 (6.2)	157.6 (5.7)	156.6 (6.1)	0.313 ^a^
Weight (kg)	52.5 (9.5)	53.6 (8.4)	54.8 (9.9)	54.6 (11.8)	0.822 ^a^	53.6 (9.2)	55.0 (9.0)	54.8 (11.7)	0.377 ^a^
Body mass index (kg/m^2^)	21.3 (3.6)	21.9 (3.1)	22.1 (4.1)	22.4 (4.1)	0.834 ^a^	21.9 (3.5)	22.2 (3.5)	22.3 (4.1)	0.801 ^a^
Waist–hip ratio	0.9(0.1)	0.88 (0.06)	0.88 (0.06)	0.87 (0.07)	0.667 ^a^	0.88 (0.06)	0.88 (0.06)	0.87 (0.07)	0.667 ^a^
Fat mass (kg)	14.6 (6.9)	15.4 (6.0)	16.2 (7.6)	16.3 (8.2)	0.898 ^a^	15.5 (6.7)	16.3 (6.7)	16.2 (8.4)	0.565 ^a^
Lean mass (kg)	38.1 (3.7)	38.2 (3.6)	38.5 (3.5)	38.3 (4.4)	0.814 ^a^	38.1 (3.7)	38.7 (3.4)	38.6 (4.3)	0.437 ^a^
Muscle mass (kg)	35.9 (3.4)	36.0 (3.3)	36.3 (3.2)	36.1 (4.0)	0.811 ^a^	35.9 (3.4)	36.5 (3.1)	36.4 (3.9)	0.436 ^a^
Water mass (kg)	27.4 (3.3)	27.5 (3.1)	27.8 (3.2)	28.0 (4.1)	0.970 ^a^	27.5 (3.3)	28.0 (3.3)	28.0 (3.7)	0.494 ^a^
Basal metabolism (MJ/day)	4.59 (0.53)	4.61 (0.50)	4.68 (0.51)	4.65 (0.64)	0.748 ^a^	4.60 (0.52)	4.70 (049)	4.69 (0.63)	0.314 ^a^
Visceral fat level	5.0 (2.7)	5.3 (2.4)	5.4 (3.0)	5.4 (3.2)	0.962 ^a^	5.3 (2.6)	5.4 (2.5)	5.2 (3.5)	0.492 ^a^
Resting energy expenditure (MJ/day)	6.82 (1.85)	6.76 (1.88)	7.04 (1.91)	7.05 (1.69)	0.498 ^a^	6.88 (1.98)	6.80 (1.55)	7.12 (1.84)	0.924 ^a^
Body temprature (°C)	36.2 (0.6)	36.1 (0.7)	36.3 (0.5)	36.2 (0.5)	0.069 ^a^	36.1 (0.7)	36.2 (0.5)	36.2 (0.5)	0.367 ^a^
Physical fitness test									
Hand–grip strength (kg)	25.8 (4.9)	25.4 (4.8)	25.3 (4.6)	24.6 (5.3)	0.718 ^a^	25.6 (4.5)	24.7 (5.1)	24.3 (5.8)	0.421 ^a^
Ruler drop test (cm)	22.5 (4.3)	22.6 (4.3)	22.7 (4.1)	23.9 (4.7)	0.184 ^a^	22.5 (4.1)	23.7 (4.1)	23.4 (5.4)	0.089 ^a^
Sit-and-reach test (cm)	36.0 (10.1)	35.9 (10.4)	35.2 (10.4	36.0 (8.9)	0.763 ^a^	36.0 (10.1)	35.6 (10.3)	34.7 (9.7)	0.658 ^a^
Cardiovascular parameters									
Systolic blood pressure (mmHg)	125.5 (18.1)	124.4 (18.0)	127.7 (18.3)	128.8 (17.6)	0.172 ^a^	126.8 (18.6)	125.5 (17.5)	124.2 (16.8)	0.666 ^a^
Diastolic blood pressure (mmHg)	75.0 (12.4)	74.0 (11.6)	76.2 (12.6)	77.3 (14.2)	0.364 ^a^	75.3 (11.5)	74.6 (13.3)	75.5 (14.7)	0.892 ^a^
Heart rate (min^−1^)	77.0 (12.7)	78.6 (12.5)	78.2 (12.9)	78.6 (13.1)	0.908 ^a^	78.3 (12.7)	80.3 (11.8)	76.0 (14.1)	0.043 ^a^
Cardio–ankle vascular index	7.50 (0.78)	7.59 (0.76)	7.53 (0.79)	7.66 (0.82)	0.710 ^a^	7.69 (0.79)	7.44 (0.73)	7.32 (0.73)	0.008 ^a^
Ankle–brachial pressure index	1.11 (0.06)	1.12 (0.06)	1.11 (0.06)	1.09 (0.07)	0.251 ^a^	1.11 (0.06)	1.11 (0.06)	1.10 (0.07)	0.663 ^a^

Values are mean (standard deviation) or percentage. ^a^ Kruskal-Wallis test, ^b^ chi-squared test.

**Table 2 nutrients-12-03437-t002:** The lifestyle and psychological characteristics of all the participants and comparison of these factors between women with mild, moderate, and severe anxiety and depressive symptoms.

Lifestyle and Psychological Characteristics	All Participants (*n* = 289)	Anxiety				Depression			
		Mild (*n* = 156)	Moderate (*n* = 83)	Severe (*n* = 50)	*p*-Value	Mild (*n* = 178)	Moderate (*n* = 74)	Severe (*n* = 37)	*p*-Value
Lifestyle factors									
Smoking (%)									
None/fewer than 20/20 or more cigarettes per day	93.1/3.1/3.8	91.0/3.9/5.1	96.4/2.4/1.2	94.0/2.0/4.0	0.563 ^b^	92.7/2.2/5.1	97.3/2.7/0	86.5/8.1/5.4	0.487 ^b^
Drinking (%)									
Never/sometimes/daily	66.8/24.6/8.7	68.6/23.7/7.7	61.5/28.9/9.6	70.0/20.0/10.0	0.739 ^b^	64.6/27.5/7.9	70.3/21.6/8.1	70.3/16.2/13.5	0.934 ^b^
Caffeine (%)									
Never/1–2 times/3 or more times per day	9.4/34.0/56.6	9.0/35.2/55.8	11.0/32.9/56.1	8.0/32.0/60.0	0.959 ^b^	10.2/32.8/57.0	5.4/39.2/55.4	13.5/29.7/56.8	0.570 ^b^
Exercise (%)									
Yes/No	49.0/51.0	58.7/41.3	42.2/57.8	30.0/70.0	<0.001 ^b^	54.8/45.2	43.2/56.8	32.4/67.6	0.024 ^b^
Menopausal Health–Related Quality of Life Questionnaire									
Somatic symptom score (0–18 points)	14.0 (4.0)	15.1 (3.9)	13.1 (3.6)	11.3 (3.7)	<0.001 ^a^	15.0 (3.8)	12.3 (3.6)	11.6 (3.7)	<0.001 ^a^
Vasomotor symptom score (0–6 points)	5.0 (2.2)	4.4 (2.0)	3.6 (2.3)	2.9 (2.0)	<0.001 ^a^	4.3 (2.1)	3.3 (2.3)	3.4 (1.9)	<0.001 ^a^
Insomnia symptom score (0–6 points)	5.0 (2.0)	4.7 (1.8)	3.6 (2.1)	2.8 (2.1)	<0.001 ^a^	4.5 (1.9)	3.8 (2.1)	2.7 (2.2)	<0.001 ^a^
Life satisfaction score (0–15 points)	8.0 (3.5)	8.8 (3.5)	6.9 (3.0)	5.6 (2.9)	<0.001 ^a^	9.2 (2.0)	6.1 (2.7)	3.9 (2.5)	<0.001 ^a^
Social involvement score (0–16 points)	9.0 (2.9)	9.7 (2.8)	9.1 (2.9)	8.5 (3.2)	0.060 ^a^	10.2 (2.6)	8.4 (2.8)	7.1 (3.2)	<0.001 ^a^
Hospital Anxiety and Depression Scale	12.0 (6.5)	8.8 (3.9)	16.6 (3.8)	22.4 (3.7)	<0.001 ^a^	9.5 (4.2)	17.7 (3.1)	23.6 (3.8)	<0.001 ^a^
Anxiety subscale score	7.0 (3.6)					5.5 (2.9)	8.9 (2.8)	10.9 (3.0)	<0.001 ^a^
Depression subscale score	6.0 (3.6)	4.5 (2.8)	7.7 (3.4)	9.9 (2.9)	<0.001 ^a^				
Psychotropic drug treatment (%)									
Yes/No	8.8/91.2	6.8/91.2	8.4/91.6	16.3/83.7	0.127 ^b^	7.4/92.6	12.7/87.3	8.1/91.9	0.416 ^b^

Values are mean (standard deviation) or percentage. ^a^ Kruskal-Wallis test, ^b^ chi-squared test.

**Table 3 nutrients-12-03437-t003:** The nutritional intakes that differed significantly between the three anxiety/depression severity groups.

Nutrients	All Participants (*n* = 289)	Anxiety				Depression			
		Mild (*n* = 156)	Moderate (*n* = 83)	Severe (*n* = 50)	*p*-Value ^a^	Mild (*n* = 178)	Moderate (*n* = 74)	Severe (*n* = 37)	*p*-Value ^a^
Total energy (MJ)	6.99 (1.98)	6.83 (1.95)	7.43 (2.07)	6.76 (1.85)	0.083	6.85 (1.90)	7.42 (2.11)	6.78 (2.03)	0.167
Protein (%E)	16.3 (3.2)	16.6 (3.4)	15.9 (2.8)	16.2 (2.9)	0.236	16.8 (3.4)	15.5 (2.3)	15.9 (3.3)	0.048
Carbohydrate (%E)	52.7 (7.9)	52.3 (7.8)	53.2 (8.5)	52.9 (7.0)	0.724	51.7 (7.9)	54.7 (6.9)	53.0 (9.0)	0.027
Soluble dietary fiber (g/MJ)	0.50 (0.17)	0.51 (0.18)	0.46 (0.13)	0.51 (0.19)	0.155	0.52 (0.17)	0.46 (0.15)	0.45 (0.15)	0.014
Insoluble dietary fiber (g/MJ)	1.35 (0.43)	1.40 (0.46)	1.25 (0.33)	1.36 (0.47)	0.082	1.41 (0.45)	1.25 (0.38)	1.24 (0.36)	0.007
Dietary fiber (g/MJ)	1.90 (0.62)	1.97 (0.66)	1.77 (0.49)	1.93 (0.68)	0.102	2.00 (0.65)	1.77 (0.57)	1.74 (0.53)	0.006
Ash (g/MJ)	2.65 (0.53)	2.73 (0.58)	2.50 (0.40)	2.64 (0.50)	0.015	2.7 (0.6)	2.5 (0.4)	2.5 (0.6)	0.005
Potassium (mg/MJ)	397.1 (109.0)	413.4 (116.2)	366.2 (82.0)	397.4 (115.7)	0.015	417.2 (109.5)	367.1 (101.5)	360.3 (99.8)	<0.001
Calcium (mg/MJ)	85.5 (29.1)	103.3 (71.3)	79.9 (25.6)	84.1 (28.8)	0.017	89.8 (29.2)	78.6 (24.8)	78.7 (33.5)	0.002
Magnesium (mg/MJ)	37.7 (8.6)	41.3 (19.0)	35.7 (7.1)	38.0 (8.7)	0.029	39.2 (8.7)	35.3 (7.6)	36.7 (8.0)	0.001
Phosphorus (mg/MJ)	150.8 (32.4)	154.4 (34.6)	144.7 (29.2)	149.3 (29.3)	0.085	155.8 (33.9)	142.2 (26.0)	143.8 (32.4)	0.008
Iron (mg/MJ)	1.16 (0.30)	1.19 (0.31)	1.09 (0.23)	1.18 (0.31)	0.056	1.21 (0.30)	1.08 (0.26)	1.10 (0.29)	0.003
Zinc (mg/MJ)	1.12 (0.18)	1.13 (0.18)	1.10 (0.17)	1.10 (0.18)	0.271	1.15 (0.18)	1.08 (0.16)	1.08 (0.17)	0.007
β–Carotene (μg/MJ)	649.1 (469.1)	706.0 (544.3)	545.8 (304.5)	642.9 (415.6)	0.042	712.5 (526.5)	561.0 (341.1)	520.0 (329.9)	0.007
Retinol equivalent (μg/MJ)	109.2 (49.9)	114.3 (54.9)	97.0 (41.7)	113.3 (43.4)	0.030	114.4 (53.4)	99.0 (40.9)	104.0 (46.7)	0.110
Vitamin D (μg/MJ)	1.94 (1.31)	2.12 (1.41)	1.72 (1.00)	1.79 (1.40)	0.044	2.07 (1.40)	1.77 (0.95)	1.69 (1.43)	0.089
α-Tocopherol (mg/MJ)	1.09 (0.29)	1.13 (0.31)	1.03 (0.24)	1.10 (0.29)	0.056	1.13 (0.30)	1.04 (0.26)	0.99 (0.28)	0.006
Vitamin K (μg/MJ)	52.9 (27.1)	55.1 (29.4)	48.2 (20.5)	54.1 (28.7)	0.394	56.3 (28.5)	46.5 (22.5)	49.8 (26.7)	0.025
Vitamin B1 (mg/MJ)	0.11 (0.02)	0.12 (0.03)	0.11 (0.02)	0.11 (0.02)	0.068	0.12 (0.02)	0.11 (0.02)	0.10 (0.02)	<0.001
Vitamin B2 (mg/MJ)	0.20 (0.05)	0.20 (0.05)	0.19 (0.04)	0.20 (0.05)	0.088	0.20 (0.05)	0.18 (0.04)	0.19 (0.06)	0.005
Niacin (mgNE/MJ)	2.43 (0.63)	2.49 (0.63)	2.29 (0.60)	2.47 (0.66)	0.060	2.52 (0.62)	2.25 (0.53)	2.35 (0.78)	0.009
Vitamin B6 (μg/MJ)	185.3 (47.6)	192.0 (50.9)	169.6 (41.5)	182.9 (50.3)	0.005	193.7 (48.3)	167.3 (45.1)	171.3 (49.9)	<0.001
Folic acid (μg/MJ)	53.2 (20.1)	55.9 (21.8)	47.8 (14.9)	53.8 (20.8)	0.016	56.8 (21.0)	47.3 (17.3)	47.7 (16.9)	<0.001
Pantothenic acid (mg/MJ)	0.94 (0.19)	0.96 (0.19)	0.91 (0.18)	0.95 (0.20)	0.077	0.97 (0.19)	0.88 (0.17)	0.93 (0.22)	0.002
Vitamin C (mg/MJ)	18.9 (8.3)	20.0 (8.8)	16.8 (6.5)	19.0 (8.6)	0.020	20.3 (8.5)	17.2 (8.0)	15.4 (5.9)	<0.001

Values are mean (standard deviation). NE—niacin equivalent, ^a^ Kruskal–Wallis test.

**Table 4 nutrients-12-03437-t004:** Associations between the daily intake of vitamin B6 (10 μg/MJ) and anxiety and depressive symptoms.

Model	Anxiety			Depressive		
	OR	95% CI	*p*-Value	OR	95% CI	*p*-Value
Model 1	0.93	0.88–0.98	0.004	0.89	0.84–0.94	<0.001
Model 2	0.93	0.88–0.99	0.016	0.91	0.85–0.97	0.002
Model 3	0.97	0.90–1.04	0.407	0.89	0.80–0.99	0.041

OR—odds ratio, CI—confidence interval. Model 1: unadjusted model. Model 2: multivariate logistic regression model, adjusted for age and menopausal status. Model 3: multivariate logistic regression model, adjusted for age, menopausal status, and the background factors related to the severity of the anxiety/depressive symptoms (insomnia and depression scores/life satisfaction, social involvement, and anxiety scores).

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
