# Peer review of "Depressive Symptoms in Middle-Aged and Elderly Women Are Associated with a Low Intake of Vitamin B6: A Cross-Sectional Study"

_nutrients, 2020, doi:10.3390/nu12113437_

Round 1

Reviewer 1 Report

The observation of a relation between intake of vitamin B6 and depressive symptoms  is interesting and in line with previous studies.

The conclusion "Higher intake of vitamin B6 could help
 relieve depressive symptoms for this population"  is not justified based on the present paper. 

The Number of data presented is overwhelming and should be reduced so that the key findings of the paper is more visible. Additional data can be shown as supplementary tables. 

Table legends should be more precise so that the tables can be understood independent of the text. Notably number of participants in each group should be indicated. 

Author Response

The observation of a relation between intake of vitamin B6 and depressive symptoms is interesting and in line with previous studies.

Thank you for your comments.

The conclusion "Higher intake of vitamin B6 could help relieve depressive symptoms for this population" is not justified based on the present paper.

We appreciate your concerns on this point.  However, we consider our conclusion correct, for the following reason. First, we investigated the daily intake of 43 nutrients; then we investigated the nutritional intake that differed significantly among the three anxiety/depression severity groups. We found the associations between the daily intake of vitamin B6 and the severity of anxiety/depression. After adjusting for background factors, only the severity of depression was significantly inversely associated with vitamin B6.

The Number of data presented is overwhelming and should be reduced so that the key findings of the paper is more visible. Additional data can be shown as supplementary tables. 

Thank you for your suggestion. In accordance with your comments, we have separated Table 1 by category (revised to Table 1 and 2). In addition, we have reduced the number of data in the Table 2 (revised to Table 3) and shown as the Table 2 as supplementary table 3.

Table legends should be more precise so that the tables can be understood independent of the text. Notably number of participants in each group should be indicated.

Thank you for your comments. I have added the number of participants in the Table 1. In addition, we have revised the table 2 (revised to Table 3) legends.

Reviewer 2 Report

Thank you for the opportunity to review this interesting paper. I have only minor suggestions for improvement.

I believe the authors should clarify their use of the term VB6. It is often associated with a specific diet which may be described by Mark Bittman as a partial vegan diet. I wasn't clear if there was any reference to this diet in the paper or whether the authors were simply grouping a subset of nutrients into a category.This may beg the question are depressed people eating less meat?

Both Tables should be in landscape layout.

Table 1 should be broken up into two tables. There are different ways to do this. One would be to have a table of Physical Characteristics of the sample and then a table of Lifestyle/QoL Characteristics. Another way to do it would be to have a table of continuous demographic measures and then have a table of categorical measures.

I believe the subheading 2.2.4. Questionnaire should be plural Questionnaires.

The authors should identify how VB6 levels may lead to depression or how depression may lead to VB6 levels. Is it higher or lower? They caution about correlation is not causation, but they should really emphasize that point throughout the discussion

Author Response

Thank you for the opportunity to review this interesting paper. I have only minor suggestions for improvement.

We appreciated your comments.

I believe the authors should clarify their use of the term VB6. It is often associated with a specific diet which may be described by Mark Bittman as a partial vegan diet. I wasn't clear if there was any reference to this diet in the paper or whether the authors were simply grouping a subset of nutrients into a category.This may beg the question are depressed people eating less meat?

You have raised an important question. We investigated the intake of nutrient factors instead of a diet. VB6 was short for “vitamin B6”. We have changed VB6 into “vitamin B6” throughout the text to avoid confusion.

Both Tables should be in landscape layout.

Thank you for your comments. We made Table 1 and 2 in landscape layout to compare background factors and dairy intake of nutrients among the severity groups.

Table 1 should be broken up into two tables. There are different ways to do this. One would be to have a table of Physical Characteristics of the sample and then a table of Lifestyle/QoL Characteristics. Another way to do it would be to have a table of continuous demographic measures and then have a table of categorical measures.

Thank you for your suggestions. We have revised the table 1 (revised to Table 1 and 2) in accordance with your comments.

I believe the subheading 2.2.4. Questionnaire should be plural Questionnaires.

Thank you for your comments. We have modified the subheading 2.2.4. in accordance with your comments.

The authors should identify how VB6 levels may lead to depression or how depression may lead to VB6 levels. Is it higher or lower? They caution about correlation is not causation, but they should really emphasize that point throughout the discussion

Thank you for your comments. This is an important query. We have added the limitations in the Discussion section to emphasize this point (p.8, lines 254255).